# EXPO: STABLE REINFORCEMENT LEARNING WITH EXPRESSIVE POLICIES

**Perry Dong**
Stanford University

**Qiyang Li**
UC Berkeley

**Dorsa Sadigh**
Stanford University

**Chelsea Finn**
Stanford University

## ABSTRACT

We study the problem of training and fine-tuning expressive policies with online reinforcement learning (RL) given an offline dataset. Training expressive policy classes with online RL present a unique challenge of stable value maximization. Unlike simpler Gaussian policies commonly used in online RL, expressive policies like diffusion and flow-matching policies are parameterized by a long denoising chain, which hinders stable gradient propagation from actions to policy parameters when optimizing against some value function. Our key insight is that we can address stable value maximization by avoiding direct optimization over value with the expressive policy and instead construct an on-the-fly RL policy to maximize Q-value. We propose **EX**pressive **P**olicy **O**ptimization (EXPO), a sample-efficient online RL algorithm that utilizes an on-the-fly policy to maximize value with two parameterized policies – a larger expressive base policy trained with a stable imitation learning objective and a light-weight Gaussian edit policy that edits the actions sampled from the base policy toward a higher value distribution. The on-the-fly policy optimizes the actions from the base policy with the learned edit policy and chooses the value maximizing action from the base and edited actions for both sampling and temporal-difference (TD) backup. Our approach yields up to 2-3x improvement in sample efficiency on average over prior methods both in the setting of fine-tuning a pretrained policy given offline data and in leveraging offline data to train online.

Code: https://github.com/pd-perry/EXPO

## 1 INTRODUCTION

Robotics has seen significant progress on challenging real-world tasks by training expressive policies on large datasets via imitation learning (Black et al., 2024). Despite promising results, imitation learning methods often struggle to achieve the high reliability and performance needed for real world use-cases, even when scaled to large datasets. Fine-tuning these policies with reinforcement learning (RL) can in principle address this problem by enabling high performance through online self-improvement. Yet, existing online reinforcement learning methods are typically designed for simple Gaussian policies (Schulman et al., 2017; Fujimoto et al., 2018) and do not effectively leverage expressive pre-trained policies, such as diffusion or flow-matching policies (Chi et al., 2023) typically used in imitation learning. Can we design an efficient and effective RL fine-tuning method for expressive policy classes?

Fine-tuning expressive policies with online RL comes with a unique challenge not present in fine-tuning simpler Gaussian policies – expressive policies like diffusion or flow-matching policies are parameterized by a long chain of denoising steps, which hinders stable gradient propagation from the action output to the policy parameters whenever we want to optimize their actions against some value functions (Ding & Jin, 2024; Park et al., 2025). In the adjacent purely offline or purely online settings, many approaches have sought to avoid the gradient propagation instability by incorporating losses at intermediate denoising steps to guide the denoising process towards high-value actions (Psenka et al., 2023; Fang et al., 2024), but it is still not obvious how to best perform stable value maximization for efficient online fine-tuning.

In this work, we make the key observation that value maximization of expressive policy classes can be made much more effective and stable by *avoiding direct optimization over value* of the expressive policy itself. Instead, we can train the base expressive policy using a stable supervised

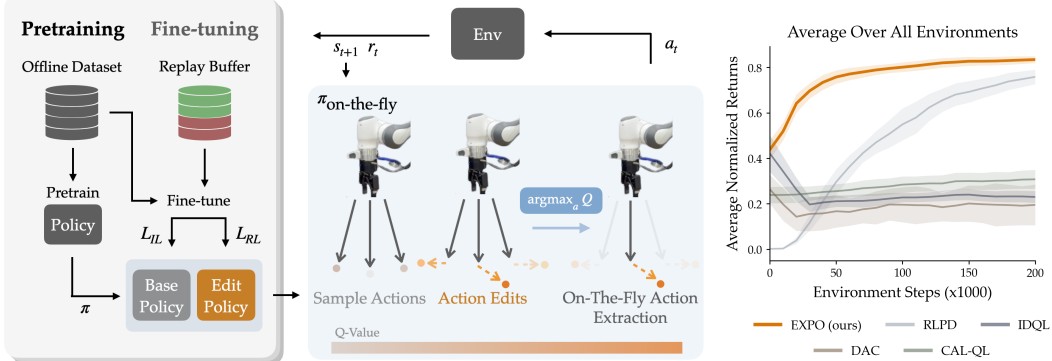

Figure 1: **Left: Expressive Policy Optimization (EXPO)** is a stable, sample efficient method for training expressive policies with reinforcement learning by avoiding direct optimization over the value function with the expressive policy. **Right: Average performance over tasks** of EXPO and prior methods.

learning objective and construct an *on-the-fly* policy to maximize value through two steps — (1) a light-weight, one-step edit policy that refines the action samples from the base expressive policy, and (2) a non-parametric post-processing step that takes multiple action candidates from the base and edit policy and selects the highest-value action among base and edited actions. We impose an *edit distance constraint* on the edit policy such that the edited actions remain close to the original actions from the base policy. This restricts the edit policy to solve a simpler, local optimization problem, allowing it to be much smaller than the base expressive policy and enabling efficient and stable optimization. The local edits can be viewed as refining actions within modes of the base policy's action distribution, which is complemented by the second on-the-fly, non-parametric post-processing step, which considers multiple pairs of base and edited actions potentially from different modes and selects the best actions.

We instantiate these insights as **EXPO**, a sample-efficient online RL algorithm that enables stable online fine-tuning of expressive policies. EXPO consists of two parameterized policies: a base expressive policy that is initialized from offline pre-training and then online fine-tuned with an *imitation learning* objective, and a small Gaussian edit policy that is trained with standard policy loss in *reinforcement learning* to maximize the $Q$-value of the edited action. The base policy is never trained to explicitly maximize value. Instead, we construct an on-the-fly policy to maximize $Q$-value by optimizing the actions from the base policy with the learned edit policy and selecting the best action from the base and edited actions according to their $Q$-values. The on-the-fly extraction has the advantage that any changes in the $Q$-function are more immediately reflected in both the agent's behavior and the TD $Q$-value target, unlike standard policy extraction methods that require slow parameter updates to align the policy to the $Q$-function. In addition, the edit policy can be trained with entropy regularization, offering a convenient way to add state-dependent action noises for online exploration beyond the behavior distribution, which is often challenging to do with expressive policies alone.

Our main contribution is a simple yet effective method for online RL fine-tuning of expressive policy classes, EXPO. Our method is stable to train and unlike many prior works that focus on a particular class of policies (e.g., diffusion, flow-matching), our method is agnostic to policy parameterization and can fine-tune from any pre-trained policies. We evaluate our method on 12 tasks across 4 domains and find that our approach achieves strong performance in both online RL and offline-to-online RL setting with up to 2-3x improvement in sample efficiency on average.

## 2 RELATED WORKS

**Reinforcement Learning with Prior Data.** To improve the sample efficiency of online RL, prior works have studied the problem of using an offline dataset to accelerate online learning (Li et al., 2023; Ball et al., 2023; Hu et al., 2023; Dong et al., 2025). A common strategy in this setting is to simply initialize the replay buffer with offline data (Vecerik et al., 2018; Nair et al., 2018; Hansen et al., 2022; Ball et al., 2023). Another line of work focuses on pretraining a good value function or policy using pessimism or policy constraints typically employed in offline RL, followed by online fine-tuning (Hester et al., 2017; Lee et al., 2021; Nair et al., 2021; Song et al., 2023; Nakamoto et al., 2024). Yet, other methods maintain separate polices for offline pretraining and online fine-tuning (Yang et al., 2023a; Zhang et al., 2023; Mark et al., 2023). However, these methods still mostly rely on simple Gaussian policies. In contrast, EXPO aims to utilize the capacity of expressive policy

classes to capture more complex behavior distributions to accelerate learning and enable fine-tuning of pre-trained models using these expressive policy classes.

**Reinforcement learning with expressive policies** started to gain popularity in RL to help handle more complex action distributions. A central focus of these methods is to extract an expressive policy that simultaneously maximizes the $Q$-function and stays close to the offline dataset. Lu et al. (2023); Kang et al. (2023); Ding et al. (2024); Zhang et al. (2025) use weighted behavior cloning (BC) to imitate dataset behavior while maximizing action $Q$-values. While weighted BC is the most simple policy extraction method that can take into account $Q$-function signals, prior works (Fu et al., 2022; Park et al., 2024; 2025) have found other policy extraction methods often performs better. Yuan et al. (2024); Ankile et al. (2024) pre-train an expressive policy on the offline data and then learn a residual policy online to refine the actions from the base policy. In contrast to these works, we focus on performing fine-tuning on the expressive policy itself, which can be crucial to fully leveraging the capabilities of the expressive policy to not only enable better sample efficiency, but also to be more adaptive online. Lastly, Ren et al. (2024) reformulate the diffusion process as an augmented MDP on top of the original MDP and use policy-gradient methods (e.g., PPO (Schulman et al., 2017)) to train the policy. Ankile et al. (2024) also uses an on-policy method to train the residual policy. Compared to these works, we focus on developing off-policy TD-based methods for better sample efficiency.

**Fine-tuning diffusion policies with value gradients.** The simplest way of leveraging the gradient of $Q$-functions for policy extraction is to backpropagate the $Q$-value into the policy parameters (Fujimoto et al., 2018; Haarnoja et al., 2019). While it is possible to directly apply this technique in diffusion policies (Wang et al., 2022), the backpropagation can get prohibitively expensive and unstable as the number of denoising steps grows large. Ding & Jin (2024); Park et al. (2025) tackles this by distilling the multi-step diffusion policy into less expressive two-step/single-step policy. Psenka et al. (2023); Fang et al. (2024) use action gradients to provide a direct supervision on the training of the intermediate denoising steps to bias towards high-value actions. Zhang et al. (2025); Mark et al. (2024) use action gradients as well but in a refinement manner where they first sample actions from the base policies and then improve these actions by hill climbing the $Q$-function using the action gradients. Our approach draws inspirations from multiple prior works, but importantly instead of backpropagating the gradient through the expressive policy, we leverage $Q$-function gradients through a separate policy to edit the base actions to maximize Q-value for better stability.

**Sampling-based maximization.** Some prior methods have explored sampling-based techniques to optimize $Q$-values. Ghasemipour et al. (2021) samples actions from the behavior policy and chooses the action that gets the highest $Q$-value and uses MADE (Germain et al., 2015) to model the behavior distribution. In contrast, we study more expressive policies for better performance. Hansen-Estruch et al. (2023) and He et al. (2024) use expressive diffusion-based policies and sampling based $Q$-function maximization only for online exploration and not TD backup. In our experiments, we find using maximum action selection for both TD backup and online exploration to be crucial for online sample efficiency. Chen et al. (2022) generalizes to softmax selection instead of a hard max for choosing actions based on the highest $Q$-values. Our method draws on ideas from these prior works, but focuses on maximizing value in a stable way to address the problem of fine-tuning expressive policies. We show through experiments not only the importance of our on-the-fly action extraction, but also editing the base actions toward higher value distributions. The design choices in our algorithm enable online RL to be more than 2x more data efficient than prior works.

## 3 PROBLEM SETTING

We consider a Markov Decision Process (MDP), defined as $\{\mathcal{S}, \mathcal{A}, r, \gamma, T, \rho\}$ where $\mathcal{S}$ is the state space, $\mathcal{A}$ is the action space, $r : \mathcal{S} \times \mathcal{A} \to \mathbb{R}$ is a function defining the rewards, $T(s'|a, s)$ is the transition dynamics, $\gamma \in [0, 1]$ is the discount factor, and $\rho(s)$ is the initial state distribution. At timestep $t$, the RL agent observes state $s_t$ and chooses action $a_t$ by sampling from its policy $\pi(a_t|s_t)$. The goal of RL is to maximize the expected sum of discounted returns $\mathbb{E}_\pi[\sum_{t=0}^{T} \gamma^t r(s_t, a_t)]$. In this paper, we study the setting where we additionally have access to a pre-trained expressive policy $\pi_{\text{pre}}$ (e.g., a diffusion policy, a flow policy) as well as a prior dataset $D_0$. As the agent interacts with the environment, it observes $(s, a, r, s')$ tuples that are appended to a replay buffer $D$ for training. Our main goal is to online fine-tune the pre-trained expressive $\pi_{\text{pre}}$ in a sample-efficient way by effectively leveraging both the prior dataset $D_0$ and the online replay buffer data $D$.

# 4 EXPRESSIVE POLICY OPTIMIZATION (EXPO)

In this section, we explain the two key components that allow EXPO to leverage a base expressive policies for sample-efficient online fine-tuning without explicitly optimizing the expressive policy for maximal rewards. The first component is an edit policy that refines the actions generated from the base policy to simultaneously maximize $Q$-value while encouraging exploratory actions. The second component is an on-the-fly policy parameterization for online training by selecting the value-maximizing action among the original and edited actions. We also present a version of EXPO with entropy backup for data-limited regimes. Lastly, we describe the implementation details required to make our method effective in practice. The full EXPO algorithm is summarized in Algorithm 1.

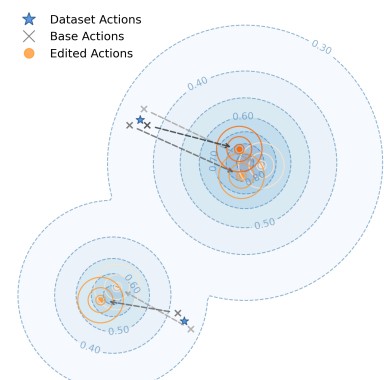

Figure 2: **The edit policy** transforms actions of the base policy into actions that further maximize $Q$-value while encouraging exploration. The blue contour represents the $Q$-values of actions of a single state and the orange contours represent the Gaussian distributions of actions the edit policy changes the base actions into.

## 4.1 Q-VALUE MAXIMIZATION AND EXPLORATION THROUGH ACTION EDITS

To avoid the unstable explicit value maximization of expressive policies, we use an imitation learning objective to train the base policy, which has been shown to work stably and reliable across a variety of expressive policy classes. However, training with imitation learning alone does not effectively move the distribution to high-value actions. To this end, the first component of EXPO is a Gaussian edit policy, $\pi_{\text{edit}}(\hat{a}|s,a)$, that refines actions generated by the base expressive policy ($a \sim \pi_{\text{base}}(\cdot|s)$):

$$\tilde{a} \leftarrow a + \hat{a} \tag{1}$$

Intuitively, we want to train the edit policy to locally optimize the $Q$-function and maximize the action entropy to maintain action diversity. Such action diversity is especially important when the base expressive policy is trained on narrow behavior distribution. We do so by training the edit policy $\pi_{\text{edit}}$ with a standard entropy-regularized policy loss:

$$L(\pi_{\text{edit}}) = -\mathbb{E}_{(s,a)\sim\mathcal{D},\hat{a}\sim\pi_{\text{edit}}(\cdot|s,a)}[Q_\phi(s, a + \hat{a}) - \alpha \log \pi_{\text{edit}}(\hat{a}|s,a)] \tag{2}$$

with $Q_\phi(s,a)$ being the critic value we want our implicit policy to maximize.

The edit policy can be viewed as transforming each action sample from the base policy toward a higher $Q$-values Gaussian action distribution. We illustrate this in Figure 2. However, naively learning this edit can shift the actions too far from the behavior distribution that it causes the policy to deviate from desirable behavior. We address this by simply enforcing the action edits to be close to the actions sampled by the policy by scaling $\hat{a}$ to be between $[-\beta, \beta]$, where $\beta$ is a hyperparameter. In practice, $\beta$ can be small (e.g., 0.05) or large (e.g., 0.7) depending on how much exploration is needed to refine the actions from the initial distribution of the offline dataset. This enables the policy to continuously improve upon the actions generated by the base policy while not deviating too far from reasonable behavior.

## 4.2 ON-THE-FLY PARAMETERIZATION OF THE RL POLICY

Given the base and edit policies, we need a way to effectively extract value-maximizing actions that account for both the expressivity of the base policy and the value-maximization of the edits. We construct an on-the-fly (OTF) policy to perform implicit value-maximization in two steps: (1) generating action samples using the base and the edit policy and (2) selecting the highest $Q$-value action. We use this on-the-fly policy for both sampling and in the TD backup.

Let $\pi_{\text{OTF}}$ be the on-the-fly policy that implicitly performs value maximization. $\pi_{\text{OTF}}(a|s, \pi_{\text{base}}, \pi_{\text{edit}}, \phi)$ is defined as $\arg\max_{a=\bigcup_{i=1}^N \{a_i, \tilde{a}_i\}} Q_\phi(s, a)$, where $a_i$ is an action sampled from $\pi_{\text{base}}$ and $\tilde{a}_i = a_i + \hat{a}_i$ is the action after edit for each of $N$ action samples. Because the edit policy is trained to maximize the $Q$-function, the edited actions should better represent what the $Q$-function views as optimal.

Taken together, the $Q$-function objective becomes

$$\min_\phi \mathbb{E}_{(s_t,a_t,s_{t+1})\sim\mathcal{D}}[(r_t + \gamma Q_{\phi'}(s_{t+1}, \tilde{a}_{t+1}^*) - Q_\phi(s_t, a_t))^2], \text{ where } \tilde{a}_{t+1}^* \sim \pi_{\text{OTF}}(\cdot|s_{t+1}) \tag{3}$$

---

**Algorithm 1** Expressive Policy Optimization (EXPO)

---

**Require:** Prior dataset $\mathcal{D}_{\text{data}} = \{(s_i, a_i)\}$; optionally, expressive policy initialization $\pi_{\text{base}}$.
    Randomly initialize action edit policy $\pi_{\text{edit}}$, critic $Q_\phi$, target critic $Q_{\phi'}$, UTD ratio $G$.
    **while** training **do**
        **for** each environment step $t$ **do**
            **Collect rollouts:**
            Sample $\tilde{a}_t^*$ from $\pi_{\text{OTF}}(\cdot|s, \pi_{\text{base}}, \pi_{\text{edit}}, \phi')$
            Take action $\tilde{a}_t^*$ and observe $r_t$ and $s_{t+1}$ from the environment
            Store $(s_t, a_t, r_t, s_{t+1})$ in RL replay buffer
            **Update policy and critic:**
            **for** $g = 1, \ldots, G$ **do**
                Sample mini-batch $(s, a, r, s')$ from the replay buffer
                Sample $\tilde{a}^{*'}$ from $\pi_{\text{OTF}}(\cdot|s', \pi_{\text{base}}, \pi_{\text{edit}}, \phi')$
                Compute target as $y = r + \gamma Q_{\phi'}(s', \tilde{a}^{*'})$
                Update $\phi$ minimizing loss: $L = (y - Q_\phi(s, a))^2$
                Update target networks: $\theta' \leftarrow \rho\theta' + (1 - \rho)\theta$
            Update $\pi_{\text{base}}$ using the last mini-batch with supervised learning objective $\mathcal{L}_{\text{IL}}(\pi_{\text{base}})$
            Update $\pi_{\text{edit}}$ using the last mini-batch maximizing objective $Q_\phi(s, a + \hat{a}) -$
        $\alpha \log \pi_{\text{edit}}(\hat{a}|s), \quad \hat{a} \sim \pi_{\text{edit}}(\cdot|s)$

---

We note that because the on-the-fly policy is parameterized to maximize the $Q$-function and the action $\tilde{a}_{t+1}^*$ is the action sample with the highest $Q$-value, this procedure can be viewed as equivalent to a standard $Q$-learning update with the implicit policy.

### 4.3 ENTROPY BACKUP FOR DATA-LIMITED REGIMES

In scenarios where the offline dataset is not sufficiently large or broad, the agent must explore more broadly during online sampling. For these scenarios, we propose a framework for incorporating an entropy bonus into both the base and edit policy training. Viewing the base and edit policies as one OTF policy, the standard entropy-regularized RL loss, which has been shown to benefit exploration (Ziebart, 2010; Haarnoja et al., 2017; 2019), can be formulated as

$$y = r_t + \gamma[Q_{\phi'}(s_{t+1}, \tilde{a}_{t+1}^*) - \alpha \log \pi_{\text{OTF}}(\tilde{a}_{t+1}^*|s_{t+1})]$$

$$L(\phi) = \mathbb{E}_{(s_t, a_t, s_{t+1}) \sim \mathcal{D}}[(y - Q_\phi(s_t, a_t))^2], \tilde{a}_{t+1}^* \sim \pi_{\text{OTF}}(\cdot|s_{t+1}) \quad (4)$$

$$L(\pi_{\text{OTF}}) = -\mathbb{E}_{(s,a) \sim \mathcal{D}, \hat{a} \sim \pi_{\text{OTF}}(\cdot|s,a)}[Q_\phi(s, a + \hat{a}) - \alpha \log \pi_{\text{OTF}}(\hat{a}|s, a)] \quad (5)$$

However, one cannot directly apply this loss to the base policy, as many expressive policies such as diffusion do not have a closed form expression for entropy. We instead construct a soft sampling distribution $\pi_{\text{sampling}}$ that first samples $N$ actions $a_i$ from $\pi_{\text{base}}$ and edits them into $\tilde{a}_{i+N} = a_i + \hat{a}_i$ for each action, and then chooses actions following the probability distribution $\pi_{\text{sampling}}(a_i|s) = \frac{\exp \beta Q(s, a_i)}{\sum_k \exp \beta Q(s, a_k)}$. We can obtain a closed-form equation for this sampling distribution and use this entropy to perform the backup. As we will see in Section 5.5, this modification can improve performance when the offline dataset is small.

### 4.4 PRACTICAL IMPLEMENTATIONS

In this paper, we instantiate EXPO with the base policy being a diffusion policy trained using DDPM. The training objective is the following:

$$\min_\psi \mathbb{E}_{t \sim \mathcal{U}(\{1, \cdots, T\}), \epsilon \sim \mathcal{N}(0, I), (s, a) \sim \mathcal{D}}[\|\epsilon - \epsilon_\psi(\sqrt{\bar{\alpha}_t}a + \sqrt{1 - \bar{\alpha}_t}\epsilon, s, t)\|]$$

While we use a diffusion policy for the main experiments as a canonical example of an expressive policy, this framework is general to any expressive policy class. We train the edit policy with a simple Gaussian with entropy regularization as done in SAC, where the entropy promotes exploration in the implicitly parameterized policy even though the base expressive policy is trained with an imitation learning objective. We turn off entropy in the target for the entropy backup version of the method.

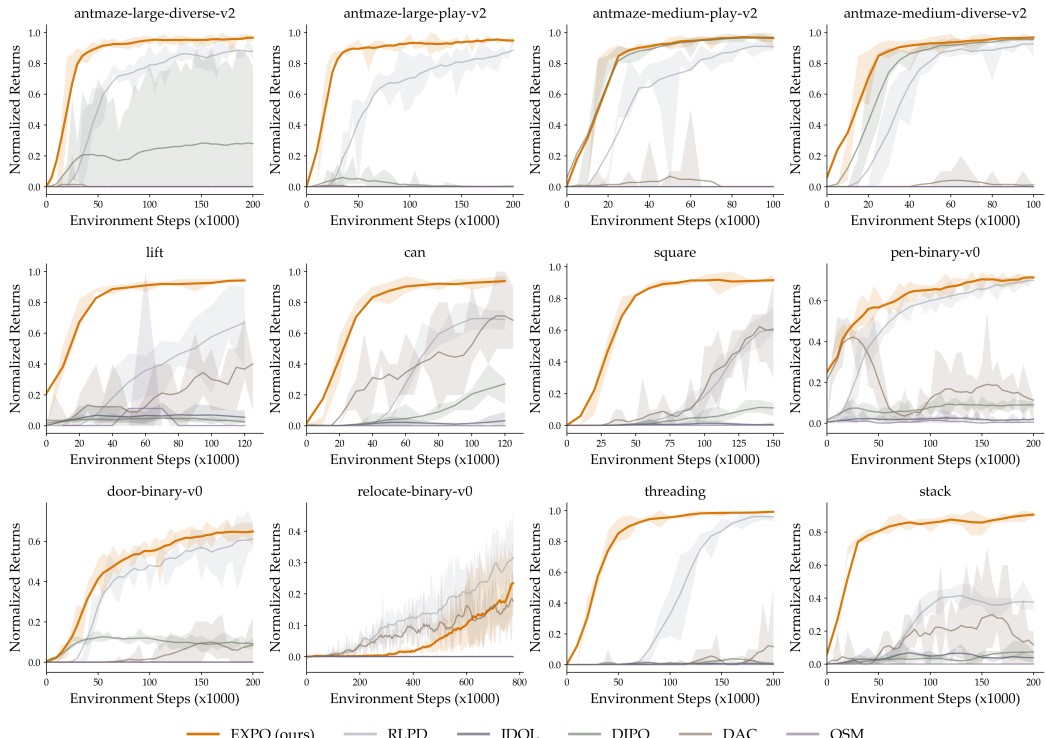

Figure 3: **Online RL results on 12 challenging sparse-reward tasks.** Across almost every task, EXPO consistently exceeds or matches the performance of the best baseline—even without any pretraining.

## 5 EXPERIMENTS

In this section, we aim to answer the following core questions through our experiments:

(**Q1**) Can EXPO effectively leverage offline data for online sample-efficient RL?

(**Q2**) How sample efficient is EXPO in fine-tuning pretrained policies compared to prior methods?

(**Q3**) What components of EXPO are most important for performance?

### 5.1 BENCHMARKS

We evaluate EXPO on 12 challenging continuous control tasks spanning various embodiments. All of the tasks feature sparse rewards. We present these tasks in Figure 11 in the Appendix. The Antmaze evaluation suite from D4RL (Fu et al., 2021) features controlling a quadruped ant to navigate a maze and reach the desired goal position. The suite consists of mazes in medium and large sizes. The Adroit environments from D4RL involves controlling a 28-Dof to spin a pen (`pen-binary-v0`), open a door (`door-binary-v0`), and relocate a ball (`relocate-binary-v0`). The RL policy needs not only to learn dexterous behavior to operate in the high-dimensional action space but also explore beyond the narrow dataset to successfully complete the tasks. The Robomimic (Mandlekar et al., 2021) and MimicGen (Mandlekar et al., 2023) tasks involve controlling a 7 DoF Franka robot arm to complete manipulation tasks. For Robomimic, we evaluate on `Lift`, `Can`, `Square`, which require lifting a block, picking a can and moving it to the correct bin, and inserting a tool onto a square peg, respectively. For MimicGen, we evaluate on `Threading` and `Stack`, which require threading a needle into a pin and stacking a small cube on top of a large cube, respectively. We initialize the dataset with successful demonstrations in all settings and tasks. We refer to the detailed setup in Section B.

### 5.2 BASELINES

We evaluate our method in both the online setting (no pre-training) as well as the offline-to-online setting (offline pre-training followed by online fine-tuning). We compare our method against prior state-of-the-art methods in each setting with a focus on methods that leverage expressive policies. As there are not many existing offline-to-online RL methods with expressive policies, we also compare to existing offline RL methods with expressive policies by directly fine-tuning them online. We describe the baselines in detail in Section C.

For the offline-to-online RL setting, we use imitation only to pre-train the base expressive policy of EXPO. This is different from other offline-to-online RL baselines such as IDQL, Cal-QL, DAC, which all use offline RL to pre-train both the policy and the value network. We only pre-train the base policy as many pre-trained robotic models do not come with a pre-trained value function. We want our method to be general and be able to fine-tune from any pre-trained policy. For Adroit, we do not pretrain for EXPO due to the narrowness of the dataset.

## 5.3 CAN EXPO EFFECTIVELY LEVERAGE OFFLINE DATA FOR ONLINE RL?

We first test whether EXPO can leverage signals from offline data of demonstrations to effectively explore and learn in an online setting. We present the results in Figure 3. We find that EXPO far exceed in performance in terms of sample efficiency compared to baselines on almost every task. Comparing against RLPD, which is a method known for its fast learning in the setting of leveraging prior data, we find that EXPO consistently achieves significantly better sample efficiency with the exception of `relocate-binary-v0` which features a very narrow dataset such that it is challenging for imitation learning to extract useful behavior from. All of this performance gain comes without pretraining on the offline data. While RLPD can learn efficiently by oversampling from the dataset, it takes a long time for the policy to discover optimal strategies, even when the information is in the offline dataset. Because EXPO is training the base policy with imitation learning, it is able to leverage signals to learn behaviors very quickly through sampling behavior close to the behavior data, and then refine those actions through the edit policy to further explore and improve in performance. Comparing against IDQL, DIPO, and QSM which uses more expressive policy classes such as diffusion, we find that these methods are often not able to learn effectively. IDQL, while also training the base policy with imitation learning and extracts actions implicitly, only does so for sampling and constrains the value function to the offline data. QSM, while in principle can learn the policy by matching the diffusion loss to action gradients, in practice often struggles to learn effectively on the challenging continuous control tasks, possibly due to instabilities in the training objective. In contrast, through a stable way of value maximization, EXPO leverages the power of expressive policy classes to achieve even better performance than simpler policy classes.

## 5.4 HOW SAMPLE EFFICIENT IS EXPO IN FINE-TUNING PRETRAINED POLICIES COMPARED TO PRIOR METHODS?

Having established the effectiveness of EXPO to leverage signals from offline datasets to effectively explore and learn, we turn our attention to the offline-to-online setting, where the policy is pretrained on the offline dataset and then finetuned. We present the results in Figure 4. EXPO achieves significantly better sample efficiency and asymptotic performance overall compared to baselines, despite only pretraining the policy using imitation learning. Crucially, compared to traditional offline-to-online RL methods, EXPO does not experience a large drop in performance from offline pretraining to online fine-tuning, despite randomly initializing both the $Q$-function and the edit policy. This is because EXPO can limit the amount of distribution shift going from offline to online as the base expressive policy generates actions that are close to the behavior distribution. While the edit policy maximizes the $Q$-value and expands the distribution to encourage exploration, it does so close to actions sampled by the base policy. Compared to IDQL with pretraining, we find that IDQL was generally not able to improve performance of the policy online after pretraining in the Antmaze and Adroit tasks, likely because of the policy constrained objective that constrains it too much to the behavior distribution combined with a lack of exploration capabilities. Cal-QL obtains strong performance on easier tasks such as `antmaze-medium-diverse-v2` and `antmaze-medium-large-v2`, but on the harder tasks has much lower overall sample efficiency despite having a calibrated $Q$-function from offline pretraining to start, as it is not able to effectively leverage signals from the offline dataset for policy improvement. DAC obtains strong pretraining performance as it takes advantage of the expressivity of diffusion models, but collapses quickly for online training, making it infeasible for fine-tuning pretrained models. With the exception of RLPD, all baselines experience an overall drop in performance going from offline to online on the Robomimic and MimicGen tasks, likely because of the precision required to complete these fine-grained manipulation tasks. In contrast, EXPO consistently improves significantly on all of the Robomimic and MimicGen tasks with high sample efficiency as the policy stays close to the behavior distribution while continuously refining the actions in a stable manner for better performance.

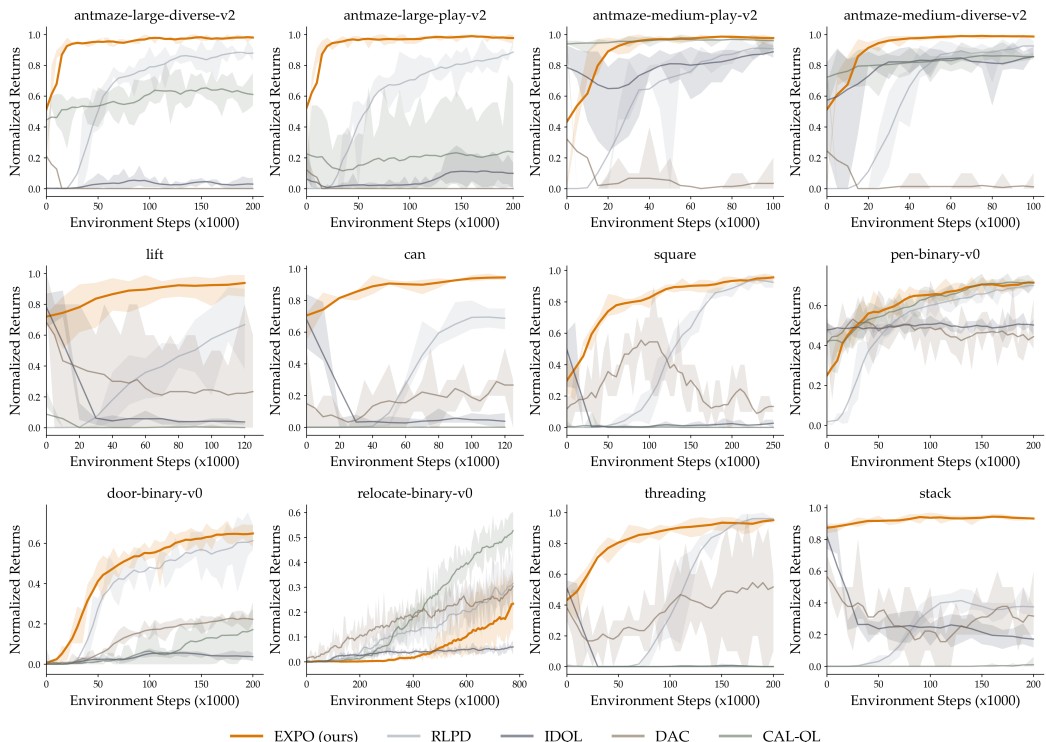

Figure 4: **Offline-to-online RL results on 12 challenging sparse-reward tasks.** EXPO consistently exceeds or matches the performance of the best baseline. The relative benefit of EXPO over baselines is especially large on the manipulation tasks, where prior methods often struggle to improve in performance. Importantly, EXPO does not drop in performance going from pre-training to fine-tuning.

## 5.5 WHAT COMPONENTS OF EXPO ARE MOST IMPORTANT FOR PERFORMANCE?

To better understand the significance of different pieces of EXPO, we ablate over three key components: (1) the importance of on-the-fly policy extraction in the TD backup, (2) the effectiveness of action edits, and (3) the importance of the behavior distribution in the offline data. We present additional experiments on fine-tuning a pre-trained policy without the offline dataset in Section A.

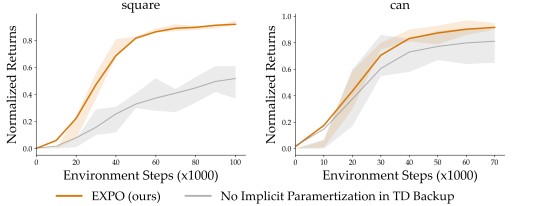

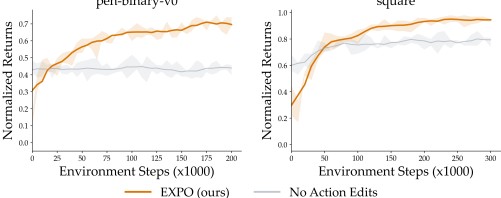

Figure 5: **Ablation over on-the-fly policy extraction in the TD backup.** We find that using value-maximizing actions in TD backup is vital for performance.

Figure 6: **Ablation over action edits.** Without action edits, it is often hard to improve pretrained policies online since the base policy by itself does not effectively explore or maximize $Q$-value.

**How important is the on-the-fly policy in TD backup?** Prior methods such as IDQL have explored sampling from an expressive imitation learning policy and choosing the highest $Q$-value action for sampling. While this parameterization is different from EXPO, the $Q$-value is also not used as a gradient signal to explicitly extract the policy. However, as the experiment results show, EXPO performs substantially better than IDQL in both online and offline-to-online settings. To better understand the role of on-the-fly value-maximization, we ablate over only performing on-the-fly action extraction for sampling, which corresponds to only sampling one action and using that action to compute the target $Q$-value, versus EXPO which extracts value maximizing actions for both sampling and backup. We present the results for Robomimic Can and Square in Figure 5. We see that on-the-fly policy extraction in the TD backup is crucial for high performance and sample efficiency. This is because while the policy is trained on implicitly maximized actions sampled in

rollout, the policy is still trained with an imitation learning objective, and as such the action sampled from the policy during TD backup does not naturally maximize the $Q$-function and thus performs a SARSA-like objective, which is known to have slower learning than $Q$-learning.

**How effective are the action edits?** To better understand the role of action edits, we compare to not using action edits and only sampling actions from the base expressive policy and choosing the action with the highest $Q$-value. We conduct the ablation on `pen-binary-v0`, an environment that requires more exploration to learn the optimal behavior, and `Square`, a task that benefits from more fine-grained refinements as the initial dataset contains useful signals to extract a behavior policy that can get a reasonable success rate. We show the results in Figure 6. The policy for `pen-binary-v0` is pretrained for 20k steps and the policy for `Square` is pretrained for 200k steps. We see that for both environments, action edits are crucial for better performance. On `pen-binary-v0`, where the policy requires more exploration, removing action edits resulted in convergence to a very suboptimal performance as the expressive policy trained with imitation learning has no mechanism to effectively explore beyond the behavior distribution. Even on `Square`, where the offline dataset contains good enough data to learn an imitation learning policy to a reasonable success rate, action edits are still very important to enable the policy to continuously refine its actions to improve.

**How does the offline dataset size affect performance with and without an entropy backup?** Because EXPO trains the base expressive policy with imitation learning, a natural question to ask is how does the offline dataset impact fine-tuning performance. To analyze the role of the offline dataset for EXPO, we subsample different number of demonstrations from the offline dataset for the `Square` task and plot the success rate of the online fine-tuned policy at 1M environment steps against the success rate of an imitation learning policy trained on the same subsampled offline dataset for both EXPO and EXPO with entropy backup. We show the results in Figure 7. We see that there is a clear pattern between fine-tuning performance and the quality of the offline dataset for EXPO without entropy, where better offline data as measured by how well an imitation learning policy trained on the data performs results in better fine-tuning performance. We note that this is perhaps not

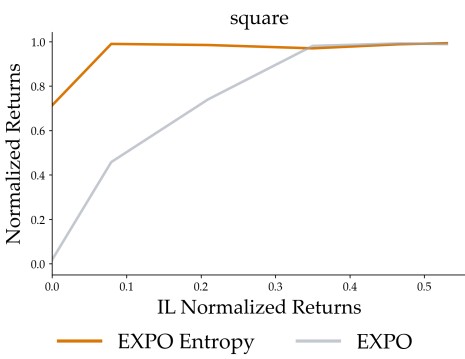

Figure 7: **Varying the offline dataset.** We find that better offline data, as measured by the performance of an imitation learning policy trained on the data, correlates strongly with performance of EXPO. The plot is averaged over 3 seeds.

surprising as both the action edits and on-the-fly value maximization rely on the assumption that the prior contains enough signals to learn useful behaviors. This also explains the lower relative sample efficiency on `relocate-binary-0`, as the offline dataset is very narrow and not sufficient for an imitation learning policy to extract useful behavior. However, EXPO with entropy backup using the soft sampling distribution described in Section 4.3 was able to address this problem by incentivizing exploration through the soft action sampling and the entropy bonus, where even an offline dataset with imitation learning performance less than 10% enabled EXPO to learn a near perfect policy. Given an offline dataset where the initial policy can learn useful behavior, we find that EXPO with or without entropy bonus consistently improves significantly over the pre-trained policy with high sample efficiency.

## 6 DISCUSSION

In this work, we propose EXPO, a method for training expressive policies with reinforcement learning given an offline dataset. Through constructing an on-the-fly RL policy using two policies, one larger expressive base policy trained with a stable imitation learning loss and one smaller edit policy trained with a Gaussian to maximize Q-value, and choosing the action generated by the policies with the highest Q-value, we address the key challenge associated with expressive policy fine-tuning, namely stable value maximization. Despite the promising results, EXPO has limitations. First, sampling many actions for the TD backup is computationally expensive, as these actions need to be sampled for every example in the batch. We leave the problem of how to improve computational efficiency for future work. Furthermore, we assume a reasonable prior either through the offline dataset or policy to start training. While in practice we believe this assumption holds in many practical settings, applying our framework to a setting with a completely uninformed prior is an interesting direction for future work.

## 7 REPRODUCIBILITY STATEMENT

For reproducibility, we describe all components of our method in detail in the main text. We include additional implementation details for hyperparameters and datasets and evaluation protocols in Section B. We also provide a link to code to run EXPO.

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

# A    ADDITIONAL EXPERIMENTS

## A.1    EXPO WITHOUT OFFLINE DATASETS

To better understand the role of the offline dataset as a prior in EXPO, we study EXPO in the setting of fine-tuning a pre-trained policy without the offline dataset used for pre-training. Instead of retaining the offline dataset, we use the pre-trained policy to collect data to warm-start the training. We present the results on Lift and Can in Figure 8 and make a comparison to Cal-QL pre-training followed by SAC fine-tuning baseline. For this ablation, we collect the

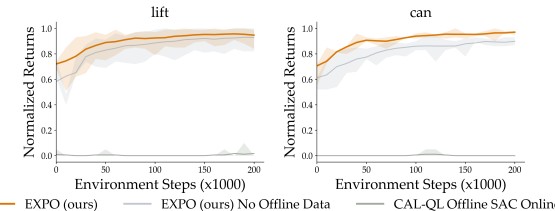

Figure 8: **Ablation on not keeping the offline dataset for fine-tuning.** We find that EXPO can learn effectively even without retaining the offline dataset after pre-training.

same number of warm-start rollouts as contained in the offline dataset used for pre-training. We find that even without retaining the offline data, EXPO was able to learn to solve the tasks with high sample efficiency similar to retaining the dataset. This is compared to Cal-QL pre-training followed by SAC finetuning, which was not able to solve the task with this setup. This suggests the pre-train policy alone can act as a strong prior for EXPO to fine-tune and improve from, and in the context of pre-trained policies, EXPO can be used for effective, sample efficient fine-tuning even without the offline dataset used to pre-train the base policy.

## A.2    ANALYSIS ON HYPERPARAMETERS

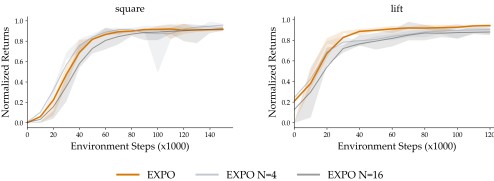

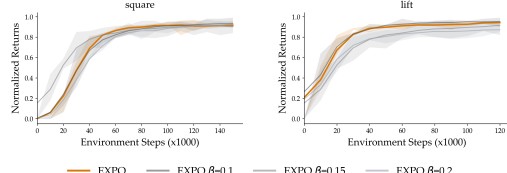

Figure 9: **Ablation over the number of samples N in the OTF policy.** We find that the number of samples N can affect performance to an extent.

Figure 10: **Ablation over the scale of action edits.** We find that the scale of actions edits affect performance, and it is important to tune this hyperparameter for each task.

We ablate over the important hyperparameters of EXPO, namely the number of samples N in the OTF policy for both sampling and TD backup and the scale of the edit policy $\beta$, to analyze the effect of these hyperparameters on performance. We present the results on Lift and Square in Figure 9 and Figure 10. We find that the performance of EXPO can be affected by both the value of N and the scale of the edit policy, where choosing the most optimal values can result in better performance. Importantly, the scale of the edit policy is a hyperparameter that is important to tune for each task. This is because a smaller edit scale will make the edit policy edit actions closer to the base actions, which is desirable if the optimal actions are close to the base actions, while a larger edit scale allows the edits to differ more, which may be better for exploration. We provide tuning strategies in Appendix Section B.

# B    EXPERIMENT DETAILS

**Hyperparameters.** Hyperparameters we used for EXPO can be found in Table 1. Each training run presented is with three seeds and error bars indicating max and min. For offline-to-online training, we present the number of pretraining steps for each suite. We do not pretrain in the online setting. We use the same residual block structure for the base policy as IDQL (Hansen-Estruch et al., 2023).

For our experiments, we find that EXPO generally works well across a fix set of hyperparameters and we only tune the edit policy $\beta$ from $[0.05, 0.1, 0.3, 0.7]$. In terms of practical hyperparameter recommendations, we recommend a smaller value of $\beta$ (e.g., 0.05 or 0.1) to start for tasks with a good offline dataset, and a larger value of $\beta$ (e.g., 0.5, 0.7) to start for tasks where it is more important to explore to find the optimal strategy. While we do not extensively tune the number of action samples N, we note that a higher number of N might work better for higher dimensional action spaces.

| Hyperparameter | Robomimic | Adroit | Antmaze | Mimicgen |
|---|---|---|---|---|
| Optimizer | | Adam | | |
| Batch Size | | 256 | | |
| Learning Rate | | 3e-4 | | |
| Discount Factor | | 0.99 | | |
| Target Network Update $\tau$ | | 0.005 | | |
| $Q$-Ensemble Size | | 10 | | |
| N Action Samples | | 8 | | |
| UTD Ratio | | 20 | | |
| Num Min $Q$ | | 2 | | |
| T | | 10 | | |
| Beta Schedule | | Variance Preserving | | |
| Base Policy MLP Hidden Dim | | 256 | | |
| Base Policy Num Residual Blocks | | 3 | | |
| Edit Policy MLP Hidden Dim | | 256 | | |
| Edit Policy MLP Hidden Layers | | 3 | | |
| Pretraining Steps | 200k | 20k | 500k | 200k |
| Edit Policy Dropout | None | 0.1 | None | None |
| Edit Policy $\beta$ Online | 0.05 | 0.7 | 0.05 | 0.05 |
| Edit Policy $\beta$ Offline-to-Online | 0.1 | 0.7 | 0.05 | 0.05 |

Table 1: **Hyperparameters for EXPO.**

**Dataset.** We list the details of the dataset used to pretrain (offline-to-online) and initialize (online) for the Robomimic and Mimicgen environments in Table 2. We subsample 10 trajectories for Lift and use the MH dataset for Can to make the tasks harder. The Adroit and Antmaze environments use the default D4RL provided datasets.

| Hyperparameter | Num Data | Composition |
|---|---|---|
| MimicGen Stack | 200 | 10 human and 190 generated by MimicGen |
| MimicGen Threading | 50 | 10 human and 40 generated by MimicGen |
| Robomimic Lift | 10 | PH |
| Robomimic Square | 200 | PH |
| Robomimic Can | 300 | MH |

Table 2: **Dataset details for Robomimic and MicmicGen environments.**

**Evaluation.** Evaluation is performed every 5k steps with 100 episodes for the Adroit and Antmaze environments and every 10k steps with 50 episodes for Robomimic and MimicGen environments. The curves are smoothed with a running average with window size 3 or 5 depending on the task for all algorithms. The shaded region is not smoothed, so the curves may appear outside of the shaded regions. For the Adroit environments, normalized return is calculated as the percentage of the total timesteps the task is considered solved. This is the same metric as RLPD (Ball et al., 2023). All tasks use a sparse binary reward indicating whether the task has been completed successfully or not.

## C  BASELINES

**IDQL** (Hansen-Estruch et al., 2023). IDQL similarly features training an expressive diffusion policy via imitation learning and sampling multiple actions and selecting the one that maximizes the $Q$-value. However, the crucial differences are: (1) IDQL only uses the implicit policy for online exploration and use implicit Q-learning loss function for the TD backup (Kostrikov et al., 2021), (2) IDQL selects actions from action candidates directly sampled from the imitation learning policy.

**RLPD** (Ball et al., 2023). RLPD is a highly sample efficient algorithm that leverages prior data and oversamples from it for learning. RLPD uses a simpler Gaussian policy and has been shown to be

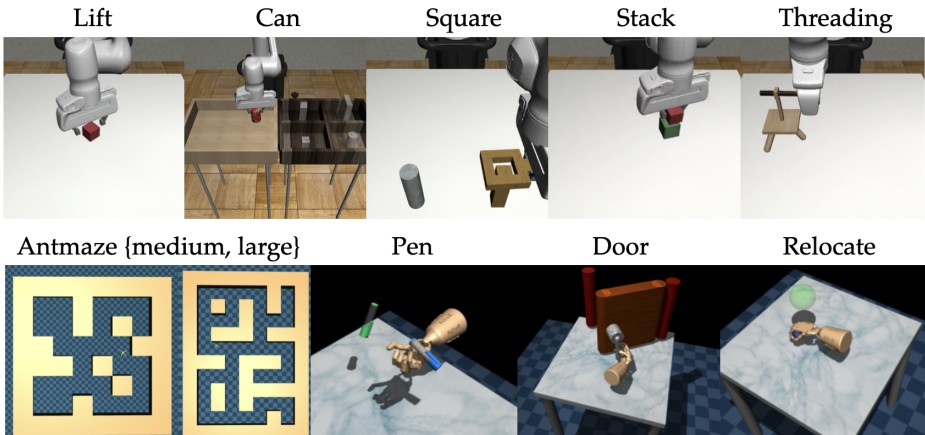

Figure 11: Visualizations of 12 sparse-reward environments we evaluate on. Note that Antmaze medium and Antmaze large both have two dataset variants.

better in performance compared to many offline-to-online methods even without pretraining. For both evaluation settings, we run RLPD without offline pre-training.

**DAC** (Fang et al., 2024). DAC is an offline RL method that uses an expressive diffusion policy. DAC includes action gradient of the $Q$-function as part of the diffusion loss to guide its denoising process towards generating more optimal actions. We adapt this method to the offline-to-online RL setting by first pre-training it with the offline RL and the continue to fine-tune it online with the same objective.

**Cal-QL** (Nakamoto et al., 2023) (**Offline-to-Online only**). Cal-QL is a standard offline-to-online RL baseline that does not use an expressive policy. Instead, Cal-QL calibrates the $Q$-function with Monte-Carlo returns as a way to balance pessimism of offline RL and optimism of online fine-tuning and prevent policy unlearning from offline to online training.

**DIPO** (Yang et al., 2023b) (**Online only**). DIPO is an online RL method based on diffusion policies. Instead of directly maximizing value through the denoising chain, DIPO performs policy improvement by using action gradient of the $Q$-function to adjust towards more optimal actions.

**QSM** (Psenka et al., 2023) (**Online only**). QSM is an online RL method that trains diffusion policies by matching the diffusion loss to action gradients. QSM aims to avoid instability of value propagation to the expressive policy by incorporating losses to guide the denoising process.

