# OpenReview forum: "EXPO: Stable Reinforcement Learning with Expressive Policies"
_ICLR.cc/2026/Conference — ICLR 2026 Poster_

### Official Review · Reviewer_RsxQ · 2025-10-31

**Soundness:** 3
**Presentation:** 3
**Contribution:** 1
**Rating:** 6
**Confidence:** 2

**Summary:**

The manuscript introduces EXpressive Policy Optimization (EXPO), a reinforcement learning (RL) algorithm designed to enable stable and sample-efficient online fine-tuning of expressive policy classes (e.g., diffusion or flow-matching policies) using both offline data and online
interaction. The key innovation is to avoid direct value maximization through the expressive policy, instead employing two parameterized components: (1) a large base expressive policy trained with an imitation learning (IL) objective, and (2) a small Gaussian edit policy that locally adjusts actions toward higher Q-values. These are combined via an on-the-fly (OTF) policy that selects the highest-value action among base and edited samples for both execution and temporal-difference (TD) backup (Sec. 4.2; Eq. (3); Fig. 2). EXPO demonstrates up to 2–3× higher sample efficiency than prior methods on 12 sparse-reward tasks across Antmaze, Adroit, Robomimic, and MimicGen domains (Fig. 3–4; Sec. 5.1–5.4).

The paper is well-motivated and tackles a clear problem—stability in RL fine-tuning of expressive policy classes—with a simple yet effective hybrid approach combining imitation learning and Q-value editing (Sec. 4.1–4.2; Fig. 1–2). Empirical coverage is broad and
convincing, with comprehensive ablations and strong baselines (Sec. 5; Fig. 3–7). The presentation is clear, though several mathematical formulations and assumptions could be better formalized (e.g., edit distance constraint, entropy-regularized variants). Moreover,
computational efficiency and theoretical analysis of stability remain underdeveloped—currently argued qualitatively (Sec. 6). Nonetheless, EXPO offers a promising direction for expressive policy fine-tuning that balances innovation with empirical rigor.

**Strengths:**

Clear motivation and problem framing:
1. The paper explicitly identifies the gradient instability of expressive policy classes (diffusion/flow policies) when directly optimizing Q-values (Sec. 1; p. 2–3). Thisconnects to recent findings on denoising-step chains (Ding & Jin 2024; Park et al.2025), establishing a solid motivation.
2. The problem is formalized as stable value maximization given a pre-trained expressive policy and dataset (Sec. 3; Eq. (1)), aligning well with practical robotic fine-tuning setups.
3. This grounding strengthens technical soundness and real-world relevance.

Elegant decomposition of learning objectives:
1. The split between base (IL-trained) and edit (Q-optimized) policies isolates instability sources and allows targeted optimization (Sec. 4.1; Eq. (2); Fig. 2).
2. The OTF mechanism (Sec. 4.2; Eq. (3)) performs implicit value maximization without direct gradient propagation through the expressive base, offering conceptual simplicity and implementation clarity.
3. This design contributes to both training stability and policy generality, as EXPO is agnostic to the expressive policy parameterization (Sec. 4.4).

Comprehensive and reproducible experimental validation:
1. Twelve challenging continuous-control tasks across four domains (Sec. 5.1; Fig. 3–4) provide strong empirical grounding.
2. Baselines include IDQL, Cal-QL, RLPD, and DAC—covering both expressive and nonexpressive methods (Sec. 5.2; App. C).
3. EXPO consistently achieves higher sample efficiency and avoids offline-to-online degradation (Fig. 4; Sec. 5.4), directly supporting the main claim.
4. Reproducibility is strengthened by hyperparameter tables (App. B; Table 1), dataset details (App. B; Table 2), and explicit evaluation protocols (App. B, p. 15).

Insightful ablations and diagnostics:
1. Ablations quantify contributions of on-the-fly TD backup (Fig. 5), edit policy (Fig. 6), and entropy backup (Sec. 5.5; Fig. 7), illustrating component necessity and interpretability.
2. Correlation between offline dataset quality and fine-tuning performance (Fig. 7) provides useful empirical evidence for practitioners.
3. The study in Appendix A demonstrates robustness even without retaining offline data, reinforcing adaptability.

Clarity and connection to prior work:
1. Related works are extensive and well-situated (Sec. 2; p. 2–4).
2. The manuscript differentiates from diffusion-based fine-tuning (DAC, QSM) and residual-policy methods (Ankile et al. 2024), improving conceptual clarity.
3. Figures 1–2 concisely visualize the EXPO workflow, enhancing accessibility.

**Weaknesses:**

Limited theoretical justification for stability and convergence:
1. The claim that separating imitation and Q-optimization yields “stable value maximization” (Sec. 4; 6) is intuitively argued but lacks formal analysis or empirical stability metrics (e.g., gradient variance, TD error oscillations).
2. No theoretical bound or convergence guarantee is provided for the coupled base-edit policy updates (Eq. (2)–(5)).
3. The assumption that local Gaussian edits suffice to capture high-value modes is unproven; sensitivity to β (edit radius) is only empirically illustrated (App. B; Table 1).
4. This limits the technical depth relative to recent theoretical RL papers.

Computational efficiency and scaling not analyzed:
1. The OTF policy requires sampling and Q-evaluation for multiple action candidates per step (Eq. (3); Sec. 4.2), which increases computational cost; yet wall-clock or FLOP comparisons are not reported (No direct evidence found in the manuscript).
2. The discussion acknowledges sampling overhead (Sec. 6 p. 9) but does not quantify trade-offs or propose scaling solutions.
3. Without resource profiling, claims of “sample efficiency” may not directly translate to runtime efficiency.

Mathematical clarity and notation consistency:
1. The use of “β” for both edit magnitude and softmax scaling (Sec. 4.1 vs. 4.3) can cause confusion; explicit notation conventions are missing.
2. The edit constraint “â ∈ [−β, β]” (Eq. (1); Sec. 4.1) lacks formal definition for vectorvalued actions—element-wise, norm-bounded, or learned scaling?
3. Entropy backup formulations (Eq. (4)–(5)) omit proofs of equivalence to standard SAC objectives under mixed sampling distributions.
These minor ambiguities hinder full reproducibility.

Empirical scope and generalization:
1. Experiments focus exclusively on robotics control benchmarks; no results on nonrobotic or discrete-action tasks (Sec. 5).
2. Although the method is claimed to be “agnostic to policy parameterization” (Sec. 6), no experiments validate this with non-diffusion expressive models.
3. Generalization to large observation spaces (e.g., visual RL) or transfer scenarios remains unexplored.

Assumption dependence and hyperparameter sensitivity
1. Performance depends heavily on the presence of “reasonable priors” via offline datasets (Sec. 6; p. 9), but quantitative thresholds for dataset adequacy are not defined.
2. The β and N hyperparameters control exploration and action sampling; only limited tuning guidelines are provided (App. B).
3. Lack of systematic sensitivity analysis may limit practical deployment.

**Questions:**

Provide theoretical and empirical stability analyses:
1. Include diagnostics of gradient norms, TD-error variance, or critic loss oscillations comparing EXPO to direct value-maximization baselines (Sec. 4.1–4.2).
2. Formalize stability claims via convergence theorems or bounded-update assumptions for coupled base/edit policies (No direct evidence found in the manuscript).
3. Report quantitative ablation of β radius vs. policy divergence to support the “local edit” assumption.

Report computational cost and scaling results:
1. Add per-step runtime, GPU hours, or FLOPs for EXPO vs. IDQL/RLPD across domains (Sec. 5.1–5.2).
2. Analyze the impact of N (number of action samples) on both performance and compute (App. B Table 1).
3. Explore efficient approximations (e.g., importance-weighted sampling or learned Qsampler) to mitigate the OTF sampling cost (Sec. 6).

Clarify mathematical notation and definitions:
1. Introduce a global notation table clarifying β (edit radius vs. temperature), πOTFentropy definitions, and vector norm conventions (Sec. 4).
2. Explicitly specify whether constraints are applied element-wise or via ℓ₂-norm bounds (Eq. (1); Sec. 4.1).
3. Expand Sec. 4.3 to show derivation of Eq. (5) from SAC objective, ensuring reproducibility.

Broaden evaluation and generalization claims:
1. Include at least one non-robotic or discrete-action task (e.g., Atari or tabular MDP) to substantiate generality beyond continuous control (Sec. 5).
2. Demonstrate compatibility with another expressive policy (e.g., flow-matching or autoregressive transformer) to validate policy-agnostic claims.
3. Provide limited visual or textual policy diagnostics (e.g., action entropy or diversity plots) for interpretability.

Quantify assumption and hyperparameter robustness:
1. Define empirical metrics for “reasonable prior” dataset quality (e.g., imitation policy success rate threshold) and report how EXPO behaves below it (Sec. 5.5; Fig. 7).
2. Conduct systematic sensitivity analysis on β and N, plotting performance variance vs. hyperparameter scaling (App. B Table 1).
3. Offer practical default ranges and scaling rules-of-thumb for new user

---

> ### Author Response · Authors · 2025-11-22
> **Official Comment by Authors**
>
> Dear Reviewer RsxQ
>
> Thank you for your thoughtful feedback and acknowledgement of our work, we address the comments below and will incorporate your feedback into the final version of the paper to enhance clarity.
>
>
> > Theoretical justification
>
> We would first like to clarify we mean stability not in the formal sense and the paper does not make formal stability claims. For other theoretical results, while we do not provide additional theoretical bounds, the policy improvement is inherited from SAC if we assume the edit policy beta can be of any value since the edit policy is conditioned on the states and base actions.
>
> > Computational cost and wall time comparisons
>
> We report wall time comparisons for EXPO, RLPD, and DIPO (new baseline based on diffusion policies) all run with l40s GPUs. We note that RLPD runs faster than both EXPO and DIPO, which is expected because RLPD uses a much faster Gaussian policy. However, the speed of DIPO and EXPO are similar, which suggests the OTF policy does not add additional overhead relative to other methods that use expressive policies.
>
>
> | Steps | EXPO (minutes) | DIPO (minutes) | RLPD (minutes) |
> |----------|----------|----------|----------|
> | 0-50k | 152 | 115.2 | 52 |
> | 50k-100k | 117| 120 | 46  |
> | 100k-150k | 118 | 121 | 44 |
> | 150k-200k | 118 | 121 | 50 |
>
>
> We do note computational efficiency as a limitation of EXPO in the limitations section compared to some of the baselines since not only does EXPO use a slow policy class as the base policy (diffusion in our experiments), but EXPO also requires sampling N actions for TD backup means every example in the batch needs N samples. While this is the case, often for practical settings data is much more expensive than compute as obtaining extra data during online RL requires human time, which is a lot more expensive than running on GPUs. With that said, computational complexity is something we hope to address in future work, and some ideas for how to address the computational cost are either using action heads such that the batch sampling can be done with a much smaller network or designing ways to parallelize across GPUs.
>
>
> >  Empirical metrics for “reasonable prior” dataset quality
>
> In Figure 7, we report a performance comparison of EXPO while varying the offline dataset as measured by the imitation policy success rate. From the plot, we can see that with a dataset with ~35% successful imitation learning policy or above, EXPO can learn a near perfect policy, so the cutoff for a reasonable prior is ~35%. With EXPO with entropy, we see that a dataset with ~10% successful imitation learning policy or above, it can learn a near perfect policy.
>
>
> > Mathematical clarity and notation consistency
>
> Thanks for pointing this out, we will make sure to clarify the mathematical notation in the main paper, for example beta used in edit radius and temperature and vector norm conventions, in the final version of the paper.
>
>
>
>
> > Hyperparameter robustness
>
> We report results for analyzing the impact of N and $\beta$ on performance for the square and lift environment. The default hyperparameters for these tasks are N=16 and $\beta$=0.05. We see that EXPO is generally robust to the hyperparameter for N and $\beta$. Additionally, we will make sure to include practical default ranges for N and $\beta$ in the final version of the paper.
>
> **Ablation over N**
>
> Lift
>
> | Steps | EXPO | EXPO (N=4) | EXPO (N=16) |
> |----------|----------|----------|----------|
> | 50k | 89.67| 80.67 | 79.11 |
> | 100k | 92.56 | 89.67 | 87.44 |
> | 150k | 94.78 | 90.44 |  88.78 |
> | 200k | 95.56 | 92.33 | 88.56 |
>
>
>
> Square
>
> | Steps | EXPO | EXPO (N=4) | EXPO (N=16) |
> |----------|----------|----------|----------|
> | 50k | 81.78| 84 | 72.78 |
> | 100k | 91.33 | 87.22 | 88.78 |
> | 150k | 92 | 95.78 | 92.28 |
> | 200k | 93.78 | 94.56 | 94 |
> | 250k | 94.78 | 96 | 94.61 |
> | 300k | 96.56 | 96.33 | 95.67 |
>
>
> **Ablation over $\beta$**
>
> Lift
>
> | Steps | EXPO | EXPO ($\beta$=0.1) | EXPO ($\beta$=0.15) | EXPO ($\beta$=0.2) |
> |----------|----------|----------|----------|----------|
> | 50k | 89.67| 91.56 | 82 | 80.56 |
> | 100k | 92.56 | 95.11 | 89.56 | 86 |
> | 150k | 94.78 | 97 | 90.67 | 89 |
> | 200k | 95.56 | 97.56 | 92.56 | 90 |
>
>
> Square
>
> | Steps | EXPO | EXPO ($\beta$=0.1) | EXPO ($\beta$=0.15) | EXPO ($\beta$=0.2) |
> |----------|----------|----------|----------|----------|
> | 50k | 81.78| 76.44 | 76.78 | 71.44 |
> | 100k | 91.33 | 91.56 | 90.56 | 88.44 |
> | 150k | 92 | 94.33 | 91.44 | 93.11 |
> | 200k | 93.78 | 94.22 | 93.89 | 93.11 |
> | 250k | 94.78 | 95.11 | 95.44 | 94.44 |
> | 300k | 96.56 | 94.89 | 95.56 | 95.44 |

---

### Official Review · Reviewer_5nRg · 2025-11-01

**Soundness:** 2
**Presentation:** 2
**Contribution:** 2
**Rating:** 4
**Confidence:** 4

**Summary:**

The paper introduces Expressive Policy Optimization (EXPO), an approach for fine-tuning large pre-trained policies (like diffusion models) using a lightweight "edit policy" layered on top of the base policy. The key idea is to freeze or avoid altering the complex base policy and instead train a small edit policy that makes local adjustments to the base policy’s output actions to maximize rewards. The edit policy is updated with a standard policy-gradient algorithm augmented with entropy regularization (similar to SAC) to encourage exploration, while a critic (Q-function) is learned via a typical off-policy TD loss. Because the base policy is an expressive model (e.g. a diffusion model) for which computing entropy or direct gradients is difficult, the authors approximate the entropy term by sampling multiple actions from the base policy and fitting a simpler surrogate distribution for the edit policy’s updates (adding some engineering overhead but keeping the base policy’s parameters fixed). Overall, this technique defers the reward optimization to the constrained edit policy rather than directly modifying the pre-trained expressive policy, thereby avoiding the unstable gradients that would result from naive end-to-end fine-tuning of the large model. Empirically, EXPO is evaluated in an offline-to-online reinforcement learning setting (starting from offline pre-training and then fine-tuning online), and it shows significantly improved policy performance and stability.

**Strengths:**

Decoupling the reward optimization from the large expressive policy by introducing a small edit Gaussian policy is a practical idea that mitigates unstable gradients when fine-tuning complex policies. By avoiding direct backpropagation through the diffusion-based base policy and instead using a separate editable layer, the approach maintains stability in training. This design helps prevent the large pre-trained model from diverging due to high-variance gradient updates.

**Weaknesses:**

The core contribution of EXPO feels incremental, essentially adding a small trainable "edit head" on top of a fixed base policy and applying well-known fine-tuning strategies. Similar architectures have appeared in prior work – for example, learning a residual policy to refine the actions of a pre-trained policy is not a new concept. The authors combine standard elements (behavior cloning for the base policy, an actor-critic with entropy regularization for the edit policy, Q-learning with TD loss, etc.), so the method comes across as a straightforward amalgamation of known techniques rather than a fundamentally new algorithm or theoretical insight.

Given that the idea is not new, a deeper understanding of the idea in diffusion RL would be useful. The authors show that it is difficult to compute entropy in the diffusion setting and propose a solution. Yet this is also new a new trick to me. Besides this, the proposed idea mainly relies on intuitive arguments without rigorous verification. As a result, the contribution may be viewed as more of an engineering solution than a novel research insight.

**Questions:**

N/A

---

> ### Author Response · Authors · 2025-11-21
> **Official Comment by Authors**
>
> Dear Reviewer 5nRG,
>
> Thank you for your thoughtful feedback for our work, we would like to address your concerns below.
>
> > Straightforward combination of ideas
>
> We agree that EXPO draws upon ideas from prior work, like all methods do. We respectfully disagree that the combination is straightforward without hindsight. The design choices in EXPO enable it to be 2x more data efficient than all prior works that we are aware of. If the combination was obvious at the time, then we would expect to see it represented in prior works and not see such a large empirical gain in performance.
>
> > A deeper understanding of the idea in diffusion RL
>
> We are not sure what is meant by a deeper understanding of the idea in diffusion RL, do you mind clarifying this?

---

### Official Review · Reviewer_53K3 · 2025-11-01

**Soundness:** 3
**Presentation:** 3
**Contribution:** 3
**Rating:** 8
**Confidence:** 4

**Summary:**

This paper proposes Expressive Policy Optimization (EXPO), an online RL algorithm that enables sample-efficient training and finetuning of expressive policies such as diffusion policies. Instead of directly updating the expressive policy to maximize the Q-value, the authors propose to train a separate Gaussian edit policy with RL, while the expressive policy itself is trained via stable imitation learning. In addition, best-of-n sampling is used in both action sampling and TD backup for faster learning. Extensive experiments demonstrate strong performance in online training and finetuning with a given offline dataset.

**Strengths:**

1. The proposed method is novel to the field, and the analysis is well-supported. By using a lightweight Gaussian policy for policy improvement and using imitation learning to train the expressive policy, the method obtains a good balance between policy expressiveness and training efficiency.
2. The experiment results are solid and well analyzed. The ablation studies focus on a few of the most important components such as on-the-fly policy extraction and action edits, which demonstrate the effect of the proposed contributions well.
3. The paper is well-written and easy to follow.

**Weaknesses:**

1. The experiments only cover online reinforcement learning with a given offline dataset, while no experiments are conducted in a purely online RL setting without a pre-collected dataset.
2. The experiment does not include several strong online diffusion-policy baselines, such as DACER [1], DIPO [2], and DPMD [3], which have been shown to perform better than or comparably to QSM on various tasks.

[1] Wang Y, Wang L, Jiang Y, et al. Diffusion actor-critic with entropy regulator[J]. Advances in Neural Information Processing Systems, 2024, 37: 54183-54204.

[2] Yang L, Huang Z, Lei F, et al. Policy representation via diffusion probability model for reinforcement learning[J]. arXiv preprint arXiv:2305.13122, 2023.

[3] Ma H, Chen T, Wang K, et al. Efficient Online Reinforcement Learning for Diffusion Policy[C]//Forty-second International Conference on Machine Learning.

**Questions:**

1. Instead of scaling the edit action to lie within the range $[-\beta, \beta]$, would a more natural approach such as enforcing a KL-divergence between the edited policy and the base policy also be applicable for keeping the edited action close to the original action samples?
2. How does the proposed method perform in purely online scenario without a given offline dataset?

---

> ### Author Response · Authors · 2025-11-21
> **Official Comment by Authors**
>
> Dear Reviewer 53K3,
>
> Thank you for your insightful feedback and appreciation of our method, experiments, and writing! We provide clarifications to your questions below and will incorporate your feedback into the paper to enhance clarity.
>
>
> > Experiments only cover online reinforcement learning with a given offline dataset
>
> We focus on the setting of reinforcement learning with prior data and design the algorithm with this in mind and assume a reasonable prior either from the policy or from the offline dataset to start. Because of this, we do not focus on experiments with online RL without a prior dataset. In Figure 7, we perform an ablation on the performance varying the quality of the offline dataset as measured by the success rate of an imitation learning policy trained on the offline dataset. Because EXPO trains the base policy with imitation learning and both the action edits and on-the-fly value maximization rely on the assumption that the base policy contains enough signals to learn useful behaviors, the performance of EXPO correlates with the quality of the offline dataset. EXPO with entropy is able to greatly alleviate this, resulting in strong performance even with a very uninformative dataset.
>
> We choose this assumption of an offline dataset since in most settings in practice (ie. robotics), it is not difficult to collect enough data such that an imitation learning can achieve a nonzero success rate, and the difficulty becomes how to improve the policy in a stable, sample efficient manner beyond the offline pretraining, which is where EXPO achieves strong performance. With that said, applying the framework of EXPO to a setting with a completely uninformed prior is an interesting direction for future work, which we discuss further in the limitations.
>
>
> > Enforcing a KL-divergence between the edited policy and the base policy
>
> We choose to scale the edit action within a range since it is a simple way to make the action edits stay close to the base policy actions, but using some form of a KL divergence instead could certainly work in theory. A potential downside is the KL divergence requires a closed form distribution, which we can only get with the entropy version of EXPO, and because the KL divergence requires the same sample space for the probability distributions and the edit policy is conditioned on the actions of the policy policy, handling this could add some complexity to the framework.
>
>
> > Experiment does not include several strong online diffusion-policy baselines
>
> We provide additional experiments for DIPO below for Lift, Can, and Square. We run it on these environments for time and compute constraints and will update the paper to include the full results on all environments in the final version.
>
> **Lift**
>
> | Steps | EXPO | DIPO |
> |----------|----------|----------|
> | 50k | 89.67| 4.44 |
> | 100k | 92.56 | 3.56 |
> | 150k | 94.78 | 5.11 |
> | 200k | 95.56 | 7.78 |
>
>
> **Can**
>
> | Steps | EXPO | DIPO |
> |----------|----------|----------|
> | 50k | 95.67| 3.78 |
> | 100k | 97.5 | 20.44 |
> | 150k | 98 | 22.67 |
> | 200k | 97.33 | 14.89 |
>
>
> **Square**
>
> | Steps | EXPO | DIPO |
> |----------|----------|----------|
> | 50k | 81.78| 0.44 |
> | 100k | 91.33 | 4.89 |
> | 150k | 92 | 8.67 |
> | 200k | 93.78 | 7.56 |
> | 250k | 94.78 | 10.44 |
> | 300k | 96.56 | 7.11 |

---

### Official Review · Reviewer_6oJM · 2025-11-01

**Soundness:** 3
**Presentation:** 3
**Contribution:** 3
**Rating:** 8
**Confidence:** 4

**Summary:**

The paper considers a scenario where one pre-trains an expressive policy on offline data via imitation learning and would like to fine tune it online. Instead of directly modifying the expressive policy—which can be rather difficulty due to architectural complexity and poor gradient propagation—they learn a Gaussian "edit" policy which is optimized to produce action residuals that better maximize value over the base, expressive policy. The final policy, denoted the on-the-fly policy, then samples the edit policy multiple times, and chooses the edit which produces the largest action-value. They then update the action-values with an approximate Q-learning update, where the max operation of the bootstrap target is approximated by the aforementioned on-the-fly procedure. Across a wide variety of domains, they evaluate their approach (EXPO) empirically in both online and offline-to-online setups.

**Strengths:**

* The use of an edit policy is elegant in that the approach is agnostic to the form of the base policy (e.g., compatible with non-parametric policies).

* The empirical performance seemed relatively consistent, both in terms of variability and how it faired relative to baseline methods.

* The paper performed a variety of ablations to justify the role of various choices in their algorithm (e.g., whether or not to re-run the on-the-fly policy for the TD target, the role of the edit policy, etc.)

**Weaknesses:**

* The evaluation only considered 3 seeds. This is reconciled by the breadth of environments and the results from taking it all together, but can limit the significance of individual plots. Further, there doesn't seem to be too much variability in EXPO's performance, perhaps due to an already really good base expressive policy?

* The appendix states that it is presenting the max and min, which measures variability and not statistical confidence. Providing a measure of confidence is crucial for making claims about performance differences between methods here.

**Questions:**

* In the ablation over action edits, the performance in "square" seemed better than many of the baselines. Is this suggesting that the offline-to-online procedure of the baselines is actively making the base policy worse here?

* Can the authors comment on the computational complexity of EXPO relative to the baselines? Running the OTF policy to compute the TD target for every sample in a mini-batch sounds considerably expensive if the action-values have to compute a forward pass for every sampled edit.

* Have the authors considered alternate forms of the OTF policy? e.g., directly performing gradient-ascent in Q(s,.) starting from the base policy's action or base policy + mean edit, or conditional cross-entropy (Lim et al., 2018), etc. There have been various proposed methods in the literature for performing this approximate max operation for continuous-action Q-learning, that—given the interpretations/explanations for the benefit of the OTF policy—I'd like to hear whether there are any thoughts or intuitions around how this might perform within EXPO?

* In some of the plots (e.g., all throughout Figure 3), the curve can be completely outside of the shaded regions. This seems like a bug, if the shaded regions supposedly represent max and min—what's going on here?

---

> ### Author Response · Authors · 2025-11-21
> **Official Comment by Authors**
>
> Dear Reviewer 6oJM,
>
> Thank you for your insightful feedback and acknowledgment of our approach, empirical results, and ablations! We provide clarifications to your questions below, which will be incorporated into the paper to enhance clarity.
>
>
> > Is the offline-to-online procedure of the baselines making the base policy worse?
>
> In our experiments, we did find that in many cases the baseline RL algorithms made the pretrained policy worse. For example, IDQL decreased in performance going from offline to online. Some of the methods also had specific offline training procedures (for example DAC), which may have lower performance compared to simply doing imitation learning.
>
> > Have the authors considered alternate forms of the OTF policy?
>
> One of our motivations behind the edit policy is to avoid the instability that may arise from action gradients, which can be seen from some of the baselines (QSM, DAC). Because of this, we choose to parameterize the edit policy with something that has been shown to work well in prior deep reinforcement learning literature, namely modeling it as a Gaussian distribution.
>
> > Curves can be completely outside of the shaded regions, what's going on here?
>
> Thanks for pointing this out, the curves are smoothed with a running average with window size 3 or 5 depending on the environment for all algorithms. The shaded region is not smoothed, so the curves may appear outside of the shaded regions. We will make sure to clarify this in the paper for the final version.
>
> > Can the authors comment on the computational complexity of EXPO relative to the baselines?
>
> Depending on the setting, EXPO can be less computationally efficient than some of the baselines since sampling N actions for TD backup means every example in the batch needs N samples, which we mention in the limitations section. While this is the case, we find empirically that using a small N (ie. 8) is enough, and often data is more expensive than compute as data requires human time. With that said, computational complexity is something we hope to address in future work, and some ideas for how to address the computational cost are either using action heads such that the batch sampling can be done with a much smaller network or designing ways to parallelize across GPUs.

---

### Comment · Area_Chair_JU66 · 2025-11-23
**Action Needed: Follow-Up Assessments Pending**

Dear Reviewers,

The authors have submitted their rebuttal, and we now require your follow-up assessments to move the decision process forward. Please review the authors’ responses and update your evaluations accordingly.

Your prompt follow-up is necessary for us to finalize the meta-review.
Kindly submit your updates as soon as possible.

Best,
Area Chair

---

### Meta-Review · Area_Chair_hGUf · 2026-01-06

**Summary:**

EXPO tackles the practical problem of fine-tuning expressive, hard-to-optimize policies (e.g., diffusion/flow) by freezing the base policy trained via imitation and learning a lightweight Gaussian “edit” policy that proposes residual action corrections; an on-the-fly (best-of-N) selection then approximates the max over actions for both execution and TD bootstrapping. Across a broad set of robotics domains in offline-to-online settings, the method shows strong and relatively consistent improvements, supported by focused ablations isolating the edit policy, the on-the-fly backup, and design choices.

**Reviewer Concerns:**

The main concerns are methodological rather than conceptual: only 3 seeds are used and variability is reported as min/max rather than statistical confidence, and the on-the-fly max approximation introduces extra compute that was not initially quantified. The authors’ responses address these points reasonably (explaining smoothing vs. unsmoothed bands, acknowledging compute overhead and arguing small N suffices, adding wall-time comparisons to relevant baselines, and clarifying that baselines can indeed degrade a strong pretrained policy), but the camera-ready should still tighten reporting (confidence intervals or rliable-style summaries) and more clearly document compute/performance trade-offs and alternative max-approximation choices.

**Reviewer Scores:**

Two reviewers are strong accept (8/10) and view the approach as novel and effective, while the more skeptical reviewers mainly argue “incremental/engineering” and request stronger statistical and compute reporting rather than disputing empirical effectiveness. With the added clarifications and additional baseline runs (e.g., DIPO) plus wall-time numbers, I would expect the borderline reviewers to move slightly upward or remain at weak accept, leaving the overall panel clearly on the accept side.

---

### Decision · Program_Chairs · 2026-01-26

Accept (Poster)